# A Polynomial Time Algorithm for Log-Concave Maximum Likelihood via Locally Exponential Families

**Brian Axelrod**
Department of Computer Science
Stanford University
baxelrod@cs.stanford.edu

**Ilias Diakonikolas**
Department of Computer Science
University of Wisconsin-Madison
ilias.diakonikolas@gmail.com

**Anastasios Sidiropoulos**
Department of Computer Science
University of Illinois at Chicago
sidiropo@gmail.com

**Alistair Stewart**
Web3 Foundation
stewart.al@gmail.com

**Gregory Valiant**
Department of Computer Science
Stanford University
gvaliant@stanford.edu

## Abstract

We consider the problem of computing the maximum likelihood multivariate log-concave distribution for a set of points. Specifically, we present an algorithm which, given $n$ points in $\mathbb{R}^d$ and an accuracy parameter $\epsilon > 0$, runs in time $\text{poly}(n, d, 1/\epsilon)$, and returns a log-concave distribution which, with high probability, has the property that the likelihood of the $n$ points under the returned distribution is at most an additive $\epsilon$ less than the maximum likelihood that could be achieved via any log-concave distribution. This is the first computationally efficient (polynomial time) algorithm for this fundamental and practically important task. Our algorithm rests on a novel connection with exponential families: the maximum likelihood log-concave distribution belongs to a class of structured distributions which, while not an exponential family, "locally" possesses key properties of exponential families. This connection then allows the problem of computing the log-concave maximum likelihood distribution to be formulated as a convex optimization problem, and solved via an approximate first-order method. Efficiently approximating the (sub) gradients of the objective function is a main technical challenge in this work.

## 1  Introduction

A distribution on $\mathbb{R}^d$ is log-concave if the logarithm of its probability density function is concave:

**Definition 1** (Log-concave Density). *A probability density function $f : \mathbb{R}^d \to \mathbb{R}_+$, $d \in \mathbb{Z}_+$, is called* log-concave *if there exists an upper semi-continuous concave function $\phi : \mathbb{R}^d \to [-\infty, \infty)$ such that $f(x) = e^{\phi(x)}$ for all $x \in \mathbb{R}^d$. We will denote by $\mathcal{F}_d$ the set of upper semi-continuous, log-concave densities with respect to the Lebesgue measure on $\mathbb{R}^d$.*

---

    This paper merges two independent works [4, 35]. Authors are in alphabetical order.

Log-concave densities form a broad nonparametric family encompassing a wide range of fundamental distributions, including the uniform, normal, exponential, logistic, extreme value, Laplace, Weibull, Gamma, Chi and Chi-Squared, and Beta distributions (see, e.g., [5]). Log-concave probability measures have been extensively investigated in several scientific disciplines, including economics, probability theory and statistics, computer science, and geometry (see, e.g., [60, 3, 54, 62, 59]). The problem of *density estimation* for log-concave distributions is of central importance in the area of non-parametric estimation (see, e.g., [62, 59, 58]) and has received significant attention during the past decade in statistics [22, 38, 36, 21, 49, 6, 45] and computer science [18, 19, 2, 15, 32, 33, 16].

One reason the class of log-concave distributions has attracted this attention, both from the theoretical and practical communities, is that log-concavity is a very natural "shape constraint," which places significantly fewer assumptions on the distribution in question than most parameterized classes of distributions. In extremely high-dimensional settings when the amount of available data is not too much larger than the dimensionality, fitting a multivariate Gaussian (or some other parametric distribution) to the data might be all one can hope to do. For many practical settings, however, the dimensionality is modest (e.g., 5-20) and the amount of data is significantly larger (e.g., hundreds of thousands or millions). In such settings, making a strong assumption on the parametric form of the underlying distribution is unnecessary—there is sufficient data to fit a significantly broader class of distributions, and log-concave distributions are one of the most natural such classes. From a practical perspective, even in the univariate setting, computing the log-concave density that maximizes the likelihood of the available data is a useful primitive, with the R implementation of Rufibach and Duembgen having over 39,000 downloads [39]. As we discuss below, the amount of data required to *learn* a log-concave distribution scales exponentially in the dimension, in contrast to most parametric classes of distributions. Nevertheless, for the many practical settings with modest dimensionality and large amounts of data, there *is* sufficient data to learn. The question now is computational: how does one compute the best-fit log-concave distribution? We focus on this algorithmic question:

*Is there an efficient algorithm to compute the log-concave MLE for datapoints in $\mathbb{R}^d$?*

Obtaining an understanding of the above algorithmic question is of interest for a number of reasons. First, the log-concave MLE is *the* prototypical statistical estimator for the class, is fully automatic (in contrast to kernel-based estimators, for example), and was very recently shown to achieve the minimax optimal sample complexity for the task of learning a log-concave distribution (up to logarithmic factors) [16, 23]. The log-concave MLE also has an intriguing geometry that is of interest from a purely theoretical standpoint [22, 57]. Developing an efficient algorithm for computing the log-concave MLE is of significant theoretical interest, and would also allow this general non-parametric class of distributions to be leveraged in the many practical settings where the dimensionality is moderate and the amount of data is large. We refer the reader to the recent survey [58] for a more thorough justification for why the log-concave MLE is a desirable distribution to compute.

## 1.1 Our Results and Techniques

The main result of this paper is the first efficient algorithm to compute the multivariate log-concave MLE. For concreteness, we formally define the log-concave MLE:

**Definition 2** (Log-concave MLE). *Let $X_1, \ldots, X_n \in \mathbb{R}^d$. The log-concave MLE, $\widehat{f}_n = \widehat{f}_n(X_1, \ldots, X_n)$, is the density $\widehat{f}_n \in \mathcal{F}_d$ which maximizes the log-likelihood $\ell(f) \overset{\text{def}}{=} \sum_{i=1}^{n} \ln(f(X_i))$ over $f \in \mathcal{F}_d$.*

As shown in [22], the log-concave MLE $\widehat{f}_n$ exists and is unique. Our main result is the first efficient algorithm to compute it up to any desired accuracy.

**Theorem 1** (Main Result). *Fix $d \in \mathbb{Z}_+$ and $0 < \epsilon, \tau < 1$. There is an algorithm that, on input any set of points $X_1, \ldots, X_n$ in $\mathbb{R}^d$, and $0 < \epsilon, \tau < 1$, runs in $\text{poly}(n, d, 1/\epsilon, \log(1/\tau))$ time and with probability at least $1 - \tau$ outputs a succinct description of a log-concave density $h^* \in \mathcal{F}_d$ such that $\ell(h^*) \geq \ell(\widehat{f}_n) - \epsilon$.*

Our algorithm does *not* require that the input points $X_1, \ldots, X_n$ in $\mathbb{R}^d$ are i.i.d. samples from a log-concave density, i.e., it efficiently solves the MLE optimization problem for any input set of points. We also note that the succinct output description of $h^*$ allows for both efficient evaluation and efficient sampling. That is, we can efficiently approximate the density at a given point (within

multiplicative accuracy), and efficient sample from a distribution that is close in total variation distance.

Recent work [16, 23] has shown that the log-concave MLE is minimax optimal, within a logarithmic factor, with respect to squared Hellinger distance. In particular, the minimax rate of convergence with $n$ samples is $\tilde{\Theta}_d\big(n^{-2/(d+1)}\big)$. Combining this sample complexity bound with our Theorem 1, we obtain the first sample near-optimal and computationally efficient *proper* learning algorithm for multivariate log-concave densities. See Theorem 4 in Appendix D.

**Technical Overview**  Here we provide an overview of our algorithmic approach. Notably, our algorithm does *not* require the assumption that the input points are samples from a log-concave distribution. It runs in $\mathrm{poly}(n, d, 1/\epsilon)$ on *any* set of input points and outputs an $\epsilon$-accurate solution to the log-concve MLE. Our algorithm proceeds by convex optimization: We formulate the problem of computing the log-concave MLE of a set of $n$ points in $\mathbb{R}^d$ as a convex optimization problem that we solve via an appropriate first-order method. It should be emphasized that one needs to overcome several non-trivial technical challenges to implement this plan.

The first difficulty lies in choosing the right (convex) formulation. Previous work [22] considered a convex formulation of the problem, though that formulation seems to inherently lead to an exponential time algorithm. Given our convex formulation, a second difficulty arises: we do not have direct access to the (sub-)gradients of the objective function and the naive algorithm to compute a subgradient at a point takes exponential time. Hence, a second challenge is how to obtain an efficient algorithm for this task. One of our main contributions is a randomized polynomial time algorithm to approximately compute a subgradient of the objective function. Our algorithm for this task leverages structural results on log-concave densities established in [16] combined with classical algorithmic results on approximating the volume of convex bodies and uniformly sampling from convex sets [48, 53, 52].

We now proceed to explain our convex optimization formulation. Our starting point is a key structural property of the log-concave MLE, shown in [22]: The logarithm of the log-concave MLE $\ln \widehat{f}_n$, is a "tent" function, whose parameters are the values $y_1, \ldots, y_n$ of the log density at the $n$ input points $x^{(1)}, \ldots, x^{(n)}$, and whose log-likelihoods correspond to polyhedra. Our conceptual contribution lies in observing that while tent distributions are not an exponential family, they "locally" retain many properties of exponential families (Definition 4). This high-level similarity can be leveraged to obtain a convex formulation of the log-concave MLE that is similar in spirit to the standard convex formulation of the exponential family MLE [61]. Specifically, we seek to maximize the log-likelihood of the probability density function obtained by normalizing the log-concave function whose logarithm is the convex hull of the log densities at the samples. This objective function is a concave function of the parameters, so we end up with a (non-differentiable) convex optimization problem. The crucial observation is that the subgradient of this objective at a given point $y$ is given by an expectation under the current hypothesis density at $y$.

Given our convex formulation, we would like to use a first-order method to efficiently find an $\epsilon$-approximate optimum. We note that the objective function is not differentiable everywhere, hence we need to work with subgradients. We show that the subgradient of the objective function is bounded in $\ell_2$-norm at each point, i.e., the objective function is Lipschitz. Another important structural result (Lemma 2) allows us to essentially restrict the domain of our optimization problem to a compact convex set of appropriately bounded diameter $D = \mathrm{poly}(n, d)$. This is crucial for us, as the diameter bound implies an upper bound on the number of iterations of a first-order method. Given the above, we can in principle use a projected subgradient method to find an approximate optimum to our optimization problem, i.e., find a log-concave density whose log-likelihood is $\epsilon$-optimal.

It remains to describe how we can efficiently compute a subgradient of our objective function. Note that the log density of our hypothesis can be considered as an unbounded convex polytope. The previous approach to calculate the subgradient in [22] relied on decomposing this polytope into faces and obtaining a closed form for the underlying integral over these faces (that gives their contribution to the subgradient). However, this convex polytope is given by $n$ vertices in $d$ dimensions, and therefore the number of its faces can be $n^{\Omega(d)}$. So, such an algorithm cannot run in polynomial time.

Instead, we note that we can use a linear program (see proof of Lemma 1) to evaluate a function proportional to the hypothesis density at a point in time polynomial in $n$ and $d$. To use this oracle for the density in order to produce samples from the hypothesis density, we use Markov Chain Monte Carlo (MCMC) methods. In particular, we use MCMC to draw samples from the uniform distribution

on super-level sets and estimate their volumes. With appropriate rejection sampling, we can use these samples to obtain samples from a distribution that is close to the hypothesis density. See Lemma 3. (We note that it does not suffice to simply run a standard log-concave density sampling technique such as hit-and-run [51]. These random walks require a hot start which is no easier than the sampling technique we propose.)

Since the subgradient of the objective can be expressed as an expectation over this density, we can use these samples to sample from a distribution whose expectation is close to a subgradient. We then use stochastic subgradient descent to find an approximately optimal solution to the convex optimization problem. The hypothesis density this method outputs has log-likelihood close to the maximum.

## 1.2 Related Work

There are two main strands of research in density estimation. The first one concerns the learnability of high-dimensional parametric distributions, e.g., mixtures of Gaussians. The sample complexity of learning parametric families is typically polynomial in the dimension and the challenge is to design computationally efficient algorithms. The second research strand — which is the focus of this paper — considers the problem of learning a probability distribution under various non-parametric assumptions on the shape of the underlying density, typically focusing on the univariate or small constant dimensional regime. There has been a long line of work in this vein within statistics since the 1950s, dating back to the pioneering work of [42] who analyzed the MLE of a univariate monotone density. Since then, shape constrained density estimation has been an active research area with a rich literature in mathematical statistics and, more recently, in computer science. The reader is referred to [10] for a summary of the early work and to [44] for a recent book on the subject.

The standard method used in statistics for density estimation problems of this form is the MLE. See [14, 55, 63, 46, 43, 11, 12, 40, 17, 7, 47, 38, 9, 41, 8, 50, 62, 21, 49, 6, 45, 16] for a partial list of works analyzing the MLE for various distribution families. During the past decade, there has been a body of algorithmic work on shape constrained density estimation in computer science with a focus on both sample and computational efficiency [24–26, 18–20, 1, 2, 29, 30, 27, 31, 33, 34]. The majority of this literature has studied the univariate (one-dimensional) setting which is by now fairly well-understood for a wide range of distributions. On the other hand, the *multivariate* setting is significantly more challenging and wide gaps in our understanding remain even for $d = 2$.

For the specific problem of learning a log-concave distribution, a line of work in statistics [22, 38, 36, 21, 6] has characterized the global consistency properties of the log-concave multivariate MLE. Regarding finite sample bounds, [49, 23] gave a sample complexity *lower bound* of $\Omega_d\left((1/\epsilon)^{(d+1)/2}\right)$ for $d \in \mathbb{Z}_+$ that holds for *any* estimator, and [49] gave a near-optimal sample complexity *upper bound* for the log-concave MLE for $d \leq 3$. [33] established the first finite sample complexity upper bound for learning multivariate log-concave densities under global loss functions. Their estimator (which is different than the MLE and seems hard to compute in multiple dimensions) learns log-concave densities on $\mathbb{R}^d$ within squared Hellinger loss $\epsilon$ with $\tilde{O}_d\left((1/\epsilon)^{(d+5)/2}\right)$ samples. [16] showed a sample complexity upper bound of $\tilde{O}_d\left((1/\epsilon)^{(d+3)/2}\right)$ for the multivariate log-concave MLE with respect to squared Hellinger loss, thus obtaining the first finite sample complexity upper bound for this estimator in dimension $d \geq 4$. Building on their techniques, this bound was subsequently improved in [23] to a near-minimax optimal bound of $\tilde{O}_d\left((1/\epsilon)^{(d+1)/2}\right)$. Alas, the computational complexity of the log-concave MLE has remained open in the multivariate case. Finally, we note that a recent work [28] obtained a non-proper estimator for multivariate log-concave densities with sample complexity $\tilde{O}_d((1/\epsilon)^{d+2})$ (i.e., at least quadratic in that of the MLE) and runtime $\tilde{O}_d((1/\epsilon)^{2d+2})$.

On the empirical side, recent work [56] proposed a non-convex optimization approach to the problem of computing the log-concave MLE, which seems to exhibit superior performance in practice in comparison to previous implementations (scaling to 6 or higher dimensions). Unfortunately, their method is of a heuristic nature, in the sense that there is no guarantee that their solution will converge to the log-concave MLE.

## 2 Preliminaries

**Notation.** We denote by $X_1, \ldots, X_n \in \mathbb{R}^d$ the sequence of samples. We denote by $S_n = \text{Conv}(\{X_i\}_{i=1}^n)$ the convex hull of $X_1, \ldots, X_n$, and by $X$ the $d \times n$ matrix with columns vec-

tors $X_1, \ldots, X_n$. We write $\mathbb{1}$ for the all-ones vector of the appropriate length. For a set $Y \subset Z$, $\mathbb{1}_Y$ denotes the indicator function for $Y$.

**Tent Densities.** We start by defining tent functions and tent densities:

**Definition 3** (Tent Function). *For $y = (y_1, \ldots, y_n) \in \mathbb{R}^n$ and a set of points $X_1, \ldots, X_n$ in $\mathbb{R}^d$, we define the tent function $h_{X,y} : \mathbb{R}^d \to \mathbb{R}$ as follows:*

$$h_{X,y}(x) = \begin{cases} \max\{z \in \mathbb{R} \text{ such that } (x, z) \in \mathrm{Conv}(\{(X_i, y_i)\}_{i=1}^n)\} & \text{if } x \in S_n \\ -\infty & \text{if } x \notin S_n \end{cases}$$

The points $(X_i, y_i)$ are referred to as *tent poles*. (See Figure 1 in appendix A for the graph of an example tent function.)

Let $p_{X,y}(x) = c \exp(h_{X,y}(x))$ with $c$ chosen such that $p_{X,y}(x)$ integrates to one. We refer to $p_{X,y}$ as a *tent density* and the corresponding distribution as a *tent distribution*. Note that the support of a *tent distribution* must be within the convex hull of $X_1, \ldots, X_n$. For the remainder of the paper, we choose a scaling such that $\mathbb{1}^T y = 0$. This scaling is arbitrary, and has no significant effect on either the algorithm or its analysis.

Tent densities are notable because they contain solutions to the log-concave MLE [22]. The solution to the log-concave MLE over $X_1, \ldots, X_n$ is always a tent density, because tent densities with tent poles $X_1, \ldots, X_n$ are the minimal log-concave functions with log densities $y_1, \ldots, y_n$ at points $X_1, \ldots, X_n$.

The algorithm which we present can be thought of as an optimization over tent functions. In Section 3.1, we will show that tent distributions retain important properties of exponential families which will be useful to establish the correctness of our algorithm.

**Regular Subdivisions.** Given a tent function $h_{X,y}$ with $h_{X,y}(X_i) = y_i$, its associated *regular subdivision* $\Delta_{X,y}$ of $X$ is a collection of subsets of $X_1, \ldots, X_n \in \mathbb{R}^d$ whose convex hulls are the regions of linearity of $h_{X,y}$. See Figure 1 in appendix A for an illustration of a tent function and its regular subdivision. We refer to these polytopes of linearity as *cells*. We say that $\Delta_{X,y}$ is a *regular triangulation* of $X$ if every cell is a $d-$dimensional simplex.

It is helpful to think of regular subdivisions in the following way: Consider the hyperplane $H$ in $\mathbb{R}^{d+1}$ obtained by fixing the last coordinate. Consider the function $h_{X,y}$ as a polytope and project each face onto $H$. Each cell is a projection of a face, and together the cells partition the convex hull of $X_1, \ldots, X_n$. Observe that regular subdivisions may vary with $y$. Figure 2 in appendix A provides one example of how changing the $y$ vector changes the regular subdivision.

For a given regular triangulation $\Delta$, the associated *consistent neighborhood* $N_\Delta$ is the set of all $y \in \mathbb{R}^n$, such that $\Delta_{X,y} = \Delta$. That is, consistent neighborhoods are the sets of parameters where the *regular triangulation* remains fixed. Note that these neighborhoods are open and their closures cover the whole space. See Figure 2 in appendix A for an example of how crossing between consistent neighborhoods results in different subdivisions. We note that for fixed $X$, when $y$ is chosen in general position, $\Delta_{X,y}$ is always a regular triangulation.

## 3 Locally Exponential Convex Programs

In this section, we lay the foundations for the algorithm presented in the next section. We present the "locally" exponential form of tent distributions and show it has the necessary properties to enable efficient computation of the log-concave MLE. Though they form a broader class of distributions, "locally" exponential distributions share some important properties of exponential families. Namely, the log-likelihood optimization is convex, and the expectation of the sufficient statistic is a subgradient. This will allows us to formulate a convex program which we will be able to solve in polynomial time.

**Definition 4.** *Let $T$ be some function (possibly parametrized by $y$) and let $q_y = \exp\left(\langle T(x), y \rangle - A(y)\right)$ be a family of probability densities parametrized by $y$ with $A(y)$ acting to normalize the density so it integrates to $1$. We say that the family $\{q_y\}$ is* locally-exponential *if the following hold: (1) $A(y)$ is convex in $y$, and (2) $\mathbb{E}_{x \sim q_y}[T(x)] \in \partial_y A(y)$.*

Note that the above definition differs from an exponential family in that for exponential families $T$ may not depend on $y$.

In this section, we derive a sufficient statistic, the *polyhedral statistic*, that shows that tent distributions are in fact locally exponential. More formally, we show:

**Lemma 1.** *For tent poles $X_1, \ldots, X_n$, there exists a function $T_{X,y} : \mathbb{R}^d \to \mathbb{R}^n$ (the polyhedral statistic) such that $p_{X,y}(x) = \exp\left(\langle T_{X,y}(x), y \rangle - A(y)\right)$ corresponds to the family of tent-distributions such that $\{p_{X,y}\}$ is locally exponential. Furthermore, $T_{X,y}$ is computable in time $\mathrm{poly}(n, d)$.*

Since we know that the log-concave MLE is a tent distribution, and all tent-distributions are log-concave, we know that the optimum of the maximum likelihood convex program in Equation (3.1) corresponds to the log-concave MLE.

$$\text{MLE of tents } = \max_y \sum_i h_{X,y}(X_i) - \log \int \exp h_{X,y}(x) dx = \max_y \quad \sum_i y_i - A(y) \quad (3.1)$$

Combining the above with the fact that the sufficient statistic allows us to compute the stochastic subgradient suggests that Algorithm 1 can compute the log-concave MLE in polynomial time.

---

**Algorithm 1** ComputeLogConcaveMLE($X_1, \ldots, X_n, \epsilon$)

---

$y \leftarrow 0; c \leftarrow 8n^2 d \log(2nd); m \leftarrow \frac{2c^2}{\epsilon^2}$

**for** $i \leftarrow 1, m$ **do**
    $\eta \leftarrow c/\sqrt{i}$
    $s \sim p_{X,y}$                                                   ▷ Using Lemma 3
    $y \leftarrow y + \eta \left(\frac{1}{n}\mathbb{1} - T_{X,y}(s)\right)$      ▷ $T$ computed via Lemma 1. $\frac{1}{n}\mathbb{1}$ follows from Equation (3.1)
    **return** $y$

---

## 3.1 The Polyhedral Sufficient Statistic

Consider a regular triangulation $\Delta$ corresponding to tent distribution parametrized by $X$ and $y$. The *polyhedral statistic* is the function

$$T_{X,y}(x) : S_n \to [0,1]^n,$$

that expresses $x$ as a convex combination of corners of the cell containing $x$ in $\Delta_y$. That is $x = XT_{X,y}(x)$ where $||T_y(x)||_1 = 1$ and $T_y(x)_i = 0$ if $X_i$ is not a corner of the cell containing $x$. The polyhedral statistic gives an alternative way of writing tent functions and tent densities:

$$h_{X,y}(x) = \langle T_y(x), y \rangle \qquad p_{X,y}(x) = \exp(\langle T_y(x), y \rangle).$$

If we restrict $y$ such that $\sum_i y_i = 0$ and define $A(y) = \log \int_x p_{X,y}(x) dx$, then we can see that for every consistent neighborhood $N_\Delta$ we have an exponential family of the form

$$\exp\left(\langle T_y(x), \theta \rangle - A(y)\right) \quad \text{for } \theta \in N_\Delta. \quad (3.2)$$

While Equation (3.2) shows how subsets of tent distributions are exponential families, it also helps highlight why tent distributions are *not* an exponential family. The sufficient statistic depends on $y$ through the regular subdivision. This means that tent distributions do not admit the same factorized form as exponential families since the sufficient statistic depends on $y$.

Note that we can use any ordering of $X_1, \ldots, X_n$ to define the polyhedral sufficient statistic everywhere including on regular subdivisions that are *not* regular triangulations. Also note that, assuming that no $X_i = X_j, i \neq j$, eliminating the last coordinate using the constraint $\mathbb{1}_n^T \theta = 0$ makes each exponential family minimal. In other words, over regions where the regular subdivision does not change (for example the consistent neighborhoods), tent distributions are minimal exponential families. This means the set of tent distribution can be seen as the finite union of a set of minimal exponential families. We refer to Equation (3.3) as the exponential form for tent densities:

$$p_{X,y}(x) = \exp\left(\langle T_{X,y}(x), y \rangle - A(y)\right) \mathbb{1}_{S_n}(x). \quad (3.3)$$

Both the polyhedral statistic and tent density queries can be computed in polynomial time with the packing linear program presented in Equation (3.4). For a point $x$, the value of $y$ yields the

log-density and the vector $\alpha$ corresponds to polyhedral statistic.

$$\max y \text{ s.t. } (x, y) = \sum_i \alpha_i (X_i, y_i), \sum_i \alpha_i = 1, \alpha_i \geq 0 \qquad (3.4)$$

Note that the above combined with tent distributions being exponential families on consistent neighborhoods gives us that the properties from Lemma 1 hold true on consistent neighborhoods. We extend the proof to the full result below.

*Proof.* Convexity follows by iteratively applying known operations that preserve convexity of a function. Since a sum of convex functions is convex (see, e.g., page 79 of [13]), it suffices to show that the function $G(y) = \ln(\int \exp(h_{X,y}(x)) \mathrm{d}x)$ is convex. Since $h_{X,y}(x)$ is a convex function of $y$, by definition, $\exp(h_{X,y}(x))$ is log-convex as a function of $y$. Since an integral of log-convex functions is log-convex (see, e.g., page 106 of [13]), it follows that $\int \exp(h_y(x)) \mathrm{d}x$ is log-convex. Therefore, $G$ is convex. We have therefore established that Equation (3.1) is convex, as desired.

$\mathbb{E}_{x \sim p_{X,y}}[T_{X,y}(x)] \in \partial_y A(y)$: Note that when $y$ is in the interior of a consistent neighborhood, the polyhedral statistic LP has a unique solution and $\mathbb{E}_{x \sim p_{X,y}}[T(x)] \in \partial_y A(y)$ (by Fact 3). When $y$ is on the boundary the solution set to the LP corresponds to the convex hull of solutions corresponding to each adjacent consistent neighborhood. This corresponds to the convex hull of limiting gradients from each neighboring consistent neighborhood and is the set of subgradients. $\qquad \square$

# 4 Algorithm and Analysis

Recall that we compute the log-concave MLE via a first-order method on the optimization formulation presented in Equation (3.1). The complete method is presented in Algorithm 1. The algorithm is based on the stochastic gradient computation presented in the previous section, a standard application of the stochastic gradient method, and a sampler that we describe later in this section. Theorem 1 follows from bounding the rate of convergence of the stochastic subgradient method and the efficiency of the sampling procedure. We outline these two components below.

## 4.1 The Stochastic Subgradient Method

Recall that algorithm 1 is simply applying the stochastic subgradient method to the following convex program with $\mathbb{1}^T y = 0$: $h(y) = \langle \frac{1}{n} \mathbb{1}_n, y \rangle - A(y)$. We require a slight strengthening of the following standard result, see, e.g., Theorem 3.4.11 in [37]:

**Fact 1.** *Let $\mathcal{C}$ be a compact convex set of diameter $\mathrm{diam}(\mathcal{C}) < \infty$. Suppose that the projections $\pi_{\mathcal{C}}$ are efficiently computable, and there exists $M < \infty$ such that for all $y \in \mathcal{C}$ we have that $\|g\|_2 \leq M$ for all stochastic subgradients. Then, after $K = \Omega \left( M \cdot \mathrm{diam}(\mathcal{C}) \log(1/\tau)/\epsilon^2 \right)$ iterations of the projected stochastic subgradient method (for appropriate step sizes), with probability at least $1 - \tau$, we have that $F\left(\bar{y}^{(K)}\right) - \min_{y \in \mathcal{C}} F(y) \leq \epsilon$, where $\bar{y}^{(K)} = (1/K) \sum_{i=1}^{K} y^{(i)}$.*

We note that Fact 1 assumes that, in each iteration, we can efficiently calculate an *unbiased* stochastic subgradient, i.e., a vector $g^{(k)}$ such that $\mathbb{E}[g^{(k)}] \in \partial_y F(y^{(k)})$. Unfortunately, this is not the case in our setting, because we can only *approximately* sample from log-concave densities. However, it is straightforward to verify that the conclusion of Fact 1 continues to hold if in each iteration we can compute a random vector $\widetilde{g}^{(k)}$ such that $\|\mathbb{E}[\widetilde{g}^{(k)}] - g^{(k)}\|_2 < \delta \stackrel{\text{def}}{=} \epsilon/(2\mathrm{diam}(\mathcal{C}))$, for some $g^{(k)} \in \partial_y F(y^{(k)})$. This slight generalization is the basic algorithm we use in our setting.

We now return to the problem at hand. We note that since $T$ represents the coefficients of a convex combination $\|T(x)\| < 1$ for all $x$, bounding $M$ by 1.

Lemma 2 will show that $\mathrm{diam}(\mathcal{C}) = O(2n^2 d \log(2nd))$. This implies that if we let $c = 8n^2 d \log(2nd)$ and run SGD for $\frac{2c^2}{\epsilon^2}$ iterations, the resulting point will have objective value within $\epsilon$ of the log-concave MLE.

**Lemma 2.** *Let $X_1, \ldots, X_n$ be a set of points in $\mathbb{R}^d$ and $\hat{f}$ be the corresponding log-concave MLE. Then, we have that $R_\infty \stackrel{\text{def}}{=} \frac{\max_{i \in [n]} \hat{f}(X_i)}{\min_{i \in [n]} \hat{f}(X_i)} \leq (2nd)^{2nd}$. Converting to an $\ell_2$ norm yields a bound on the diameter of $\mathcal{C}$: $\mathrm{diam}(\mathcal{C}) \leq 2n^2 d \log(2nd)$.*

Let us briefly sketch the proof of Lemma 2. The main idea is to show that if $R_\infty$ were too high, then $\widehat{f}_n$ would have a lower likelihood than the uniform distribution on the convex hull of the samples $S_n$. More specifically, if the maximum value $M$ of the density $\widehat{f}_n$ is large, then the volume of the set $\{x \in \mathbb{R}^d : \widehat{f}_n(x) \geq M/R\}$ is small. For a fixed $R$, this set contains $S_n$ and thus $R_\infty$ must be large compared to $M\mathrm{vol}(S_n)$. Since $\widehat{f}_n$ has likelihood at least as high as the uniform distribution over $S_n$, $R$ must be small compared to $M\mathrm{vol}(S_n)$. Combining these two observations yields a bound on $R$.

We now proceed with the complete proof.

*Proof of Lemma 2.* Let $V = \mathrm{vol}(S_n)$ be the volume of the convex hull of the sample points and $M = \max_x \widehat{f}_n(x)$ be the maximum pdf value of the MLE. By basic properties of the log-concave MLE (see, e.g., Theorem 2 of [22]), we have that $\widehat{f}_n(x) > 0$ for all $x \in S_n$ and $\widehat{f}_n(x) = 0$ for all $x \notin S_n$. Moreover, by the definition of a tent function, it follows that $\widehat{f}_n$ attains its global maximum value and its global non-zero positive value in one of the points $X_i$.

We can assume without loss of generality that $\widehat{f}_n$ is not the uniform distribution on $S_n$, since otherwise $R_\infty = 1$ and the lemma follows. Under this assumption, we have that $R_\infty > 1$ or $\ln R_\infty > 0$, which implies that $M > 1/V$. The following fact bounds the volume of upper level sets of any log-concave density:

**Fact 2** (see, e.g., Lemma 8 in [16]). *Let $f \in \mathcal{F}_d$ with maximum value $M_f$. Then for all $w > 0$, we have $\mathrm{vol}(L_f(M_f e^{-w})) \leq w^d/M_f$.*

By Fact 2 applied to the MLE $\widehat{f}_n$, for $w = \ln R_\infty$, we get that $\mathrm{vol}(L_{\widehat{f}_n}(M/R_\infty)) \leq (\ln R_\infty)^d/M$. Since the pdf value of $\widehat{f}_n$ at any point in the convex hull $S_n$ is at least that of the smallest sample point $X_i$, i.e., $M/R_\infty$, it follows that $S_n$ is contained in $L_{\widehat{f}_n}(M/R_\infty)$. Therefore, $V \leq (\ln R_\infty)^d/M$.

On the other hand, the log-likelihood of $\widehat{f}_n$ is at least the log-likelihood of the uniform distribution $U_{S_n}$ on $S_n$. Since at least one sample point $X_i$ has pdf value $\widehat{f}_n(X_i) = M/R_\infty$ and the other $n-1$ sample points have pdf value $\widehat{f}_n(X_i) \leq M$, we have that $\ln(M/R_\infty) + (n-1)\ln M \geq \ell(\widehat{f}_n) \geq \ell(U_{S_n}) = n\ln(1/V)$, or $n\ln M - \ln R_\infty \geq -n\ln V$, and therefore $\ln(MV) \geq (\ln R_\infty)/n$. This gives that $R_\infty^{1/n} \leq MV$. Combining this expression with $V \leq (\ln R_\infty)^d/M$ from above yields that $R_\infty \leq (\ln R_\infty)^{nd}$.

Since $\ln x < x$, $x \in \mathbb{R}$, setting $x = R_\infty^{\frac{1}{2nd}}$ gives that $\ln R_\infty < 2nd \cdot R_\infty^{\frac{1}{2nd}}$ or $(\ln R_\infty)^{nd} < (2nd)^{nd} \cdot R_\infty^{1/2}$. By the above, we deduce that $R_\infty \leq (2nd)^{nd} \cdot R_\infty^{1/2}$ or $R_\infty \leq (2nd)^{2nd}$. This completes the proof of Lemma 2. $\qquad\square$

## 4.2 Efficient Sampling and Log-Partition Function Evaluation

In this section, we establish the following result, which gives an efficient algorithm for sampling from the log-concave distribution computed by our algorithm.

**Lemma 3** (Efficient Sampling). *There exist algorithms $\mathcal{A}_1$ and $\mathcal{A}_2$ satisfying the following: Let $\delta, \tau > 0$, let $X = X_1, \ldots, X_n \in \mathbb{R}^d$, let $y \in \mathbb{R}^n$ be a parameter of a tent-density in exponential form. Then the following conditions hold:*

(1) *On input $X$, $y$, $\delta$, and $\tau$, algorithm $\mathcal{A}_1$ outputs a random vector $Z \in \mathbb{R}^d$, distributed according to some probability distribution with density $\widetilde{\phi}$, such that $\|\widetilde{\phi} - p_{X,y}\|_1 = O(\delta)$, in time $\mathrm{poly}(n, d, \|y\|_\infty, 1/\delta, \log(1/\tau))$, with probability at least $1 - \tau$.*

(2) *On input $X$, $y$, $\delta$, and $\tau$, algorithm $\mathcal{A}_2$ outputs some $\gamma' > 0$, such that $\gamma'/(1 + O(\delta)) \leq \int \exp(h_{X,y}(x))\mathrm{d}x \leq \gamma' \cdot (1 + O(\delta))$, in time $\mathrm{poly}(n, d, \|y\|_\infty, 1/\delta, \log(1/\tau))$, with probability at least $1 - \tau$.*

The algorithm used to show lemma 3 is presented in algorithm 2.

Algorithm 2 operates in two stages (The formal analysis is presented in appendix B.) First, it slices the tent distribution into level sets and computes their volume. Properties of log-concave distributions

---
**Algorithm 2** Algorithm to sample from $p_{X,y}$
---
**procedure** SAMPLE($X_1, \ldots, X_n, y$)
**Input:** Sequence of points $X = \{X_i\}_{i=1}^n$ in $\mathbb{R}^d$, vector $y \in \mathbb{R}^n$, parameter $0 < \delta < 1$.
**Output:** A random vector $Z \in \mathbb{R}^d$ sampled from a probability distribution with density function $\widetilde{\phi}$, such that $\|\widetilde{\phi} - p_{X,y}\|_1 \leq \delta$.
**Step 1.** Let $m = \lceil 1 + 2\|y\|_\infty \rceil$. Let $M = \max_{x \in \mathbb{R}^d} \exp(h_{X,y}(x))$. For any $i \in [m]$, let $L_i = \{x \in \mathbb{R}^d : \exp(h_{X,y}(x)) \geq M \cdot 2^{-i}\}$. For each $i \in [m]$ compute an estimate $\widetilde{\mathrm{vol}}(L_i)$ of $\mathrm{vol}(L_i)$ such that

$$\mathrm{vol}(L_i)/(1+\delta) \leq \widetilde{\mathrm{vol}}(L_i) \leq \mathrm{vol}(L_i)(1+\delta).$$

**Step 2.** For $i \in [m]$, let $u_i$ be the uniform probability distribution on $L_i$, and let $\widetilde{u}_i$ be an efficiently samplable probability distribution such that

$$\|\widetilde{u}_i - u_i\|_1 \leq \delta.$$

**Step 3.** Let $\widetilde{c} = \sum_{i=1}^m 2^{-i} \widetilde{\mathrm{vol}}(L_i) + 2^{-m} \widetilde{\mathrm{vol}}(L_m)$.
**Step 4.** Let $\widehat{D}$ be the probability distribution on $[m]$ with

$$\Pr_{I \sim \widetilde{D}}[I = i] = \begin{cases} \widetilde{\mathrm{vol}}(L_i) \cdot 2^{-i}/\widetilde{c} & \text{if } i \in \{1, \ldots, m-1\} \\ 2 \cdot \widetilde{\mathrm{vol}}(L_m) \cdot 2^{-m}/\widetilde{c} & \text{if } i = m \end{cases}$$

**Step 5.** Sample $I \sim \widetilde{D}$ and sample $Z \sim \widetilde{u}_I$.
**Step 6.** For any $x \in \mathbb{R}^d$ let $G_{X,y}(x) = M \cdot 2^{-\lfloor \log_2(M/\exp(h_{X,y}(x))) \rfloor}$.
**Step 7.** With probability $1 - \exp(h_{X,y}(Z))/G_{X,y}(Z)$ go to Step 5.
**return** $Z$.
---

allow us to guarantee that we obtain a good approximation of the the density with these slices. We then derive a linear program which can be used as a separation oracle for tent densities. This allows us to compute their volume using a classic result for volume estimation [48]. In the second stage we sample from the "sliced" distribution above. We first draw a single random number to choose a level set, weighted by the volume computed in the first stage. We then draw a sample uniformly at random from the corresponding level set and return that as our sample. Please see appendix B for a complete exposition, proof and pseudocode.

## 5 Conclusions

In this paper, we gave a $\mathrm{poly}(n, d, 1/\epsilon)$ time algorithm to compute an $\epsilon$-approximation of the log-concave MLE based on $n$ points in $\mathbb{R}^d$. Ours is the first algorithm for this problem with a sub-exponential dependence in the dimension $d$. We hope that our approach may lead to more practical methods for computing the log-concave MLE in higher dimensions than was previously possible.

One concrete open question is whether there exists an algorithm for computing the log-concave MLE that runs in time $\mathrm{poly}(n, d, \log(1/\epsilon))$, instead of the $\mathrm{poly}(n, d, 1/\epsilon)$ that we achieve. Such an algorithm would likely be technically interesting as it may require going beyond the first-order methods we employ. More broadly, it seems worth investigating whether the MLE can be efficiently computed for other natural classes of non-parametric distributions. Alternately, one could hope that there is a simple set of natural properties such that, if a class of distributions satisfies those properties, then the MLE can be efficiently computed.

**Acknowledgments:** Ilias Diakonikolas was supported by NSF Award CCF-1652862 (CAREER) and a Sloan Research Fellowship. Alistair Stewart was supported by a USC startup grant. Anastasios Sidiropoulos was supported by NSF awards CCF-1453472 (CAREER) and CCF-1934915 (TRIPODS), and NSF grants CCF-1423230 and CCF-1815145. Brian Axelrod was supported by NSF Fellowship grant DGE-1656518, NSF award CCF-1763299, and a Finch family fellowship. Brian Axelrod and Gregory Valiant were supported by NSF awards CCF-1704417, and an ONR Young Investigator Award (N00014-18-1-2295).

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
