[Supplementary Material]

# Appendix

## A Figures

Figure 1: An example of a tent function and its corresponding regular subdivision. Notice that the regular subdivision is *not* a regular triangulation.

Figure 2: Changing the height of the tent poles can change the induced regular subdivision (shown in purple).

## B Sampling Algorithm

The algorithm used in the proof of Lemma 3 is concerned mainly with part (1) in its statement. The pseudocode of this sampling procedure is given in Algorithm 2.

Using the notation from Algorithm 2, part (2) is easier to describe and we thus omit the pseudocode. We note that the following exposition of Algorithm 2 assumes that the input vector $y$ is bounded. In the execution of Algorithm 1, $\|y\|_\infty$ is bounded linearly by the number of SGD iterates. Thus, the dependence of the sampling runtime on $\|y\|_\infty$ increases the overall runtime by at most a polynomial.

We now present the proof of Lemma 3. The pseudocode of the sampling procedure is given in Algorithm 2. As stated in Section 4.1.2, Algorithm 2 uses subroutines for approximating the volume of a convex body given by a membership oracle, and a procedure for sampling from the uniform distribution supported on such a body. For these procedures we use the algorithms by [48], which are summarized in Theorems 2 and 3 respectively.

**Theorem 2** ([48]). *The volume of a convex body $K$ in $\mathbb{R}^d$, given by a membership oracle, can be approximated to within a relative error of $\delta$ with probability $1 - \tau$ using*

$$d^5 \cdot \mathrm{poly}(\log d, 1/\delta, \log(1/\tau))$$

*oracle calls.*

**Theorem 3** ([48]). *Given a convex body $K \subset \mathbb{R}^d$, with oracle access, and some $\delta > 0$, we can generate a random point $u \in K$ that is distributed according to a distribution that is at most $\delta$ away from uniform in total variation distance, using*

$$d^5 \cdot \mathrm{poly}(\log d, 1/\delta)$$

*oracle calls.*

For all $X = X_1, \ldots, X_n \in \mathbb{R}^d$, $y \in \mathbb{R}^n$, and $x \in \mathbb{R}^d$, we use the notation $H_{X,y}(x) = \exp(h_{X,y}(x))$.

In order to use the algorithms in Theorems 2 and 3 in our setting, we need a membership oracle for the superlevel sets of the function $H_{X,y}$. Such an oracle can clearly be implemented using the LP (3.4). We also need a separation oracle for these superlevel sets, which is given in the following lemma:

**Lemma 4** (Efficient Separation). *There exists a $\mathrm{poly}(n, d)$ time separation oracle for the superlevel sets of $H_{X,y}(x) = \exp(h_{X,y}(x))$.*

*Proof.* To construct our separation oracle, we will rely on the covering LP that is dual to the packing LP used to evaluate a tent function. The dual to the packing LP looks for the hyperplane that is above all the $(X_i, y_i)$ that has minimal $y$ at $x$. More specifically, it is the following LP:

$$\begin{aligned} \text{minimize} \quad & \beta_0 + \sum_{j=1}^d \beta_j x_j \\ \text{subject to} \quad & \beta \in \mathbb{R}^{d+1}, \beta_0 + \sum_{j=1}^d \beta_j X_{i,j} \geq y_i, i \in [n] \,, \end{aligned} \tag{B.1}$$

where $X_{i,j}$ is the $j$-th coordinate of the vector $X_i$. Now suppose that we are interested in a super level set $L_{H_{X,y}}(l)$. We can use the above LP to compute $h_{X,y}(x)$ (and thus $H_{X,y}(x)$) and check if it is in the superlevel set. Suppose that it is not, then there will be a solution $\beta \in \mathbb{R}^{d+1}$ whose value is below $\ln l$, say $\ln l - \delta$ for some $\delta > 0$. Consider an $x'$ in the halfspace $\beta_0 + \sum_{j=1}^d \beta_j x'_j \leq \ln l - \delta/2$ which has $x$ in the interior. Since $x$ does not appear in the objective, $\beta$ is a feasible solution for the dual LP (B.1) with $y, x'$, and so $h_y(x') \leq \ln l - \delta/2$, which implies that $x'$ is not in the superlevel set. Therefore, $\beta_0 + \sum_j \beta_j x'_j = \ln l - \delta/2$ is a separating hyperplane for $x$ and the level set. This completes the proof. $\qquad\square$

Given all of the above ingredients, we are now ready to prove the main result of this section.

*Proof of Lemma 3.* We first prove part (1) of the assertion. To that end we analyze the sampling procedure described in Algorithm 2. Recall that $m = 1 + \lceil \|y\|_\infty \rceil$, and for any $i \in [m]$, we define the superlevel set

$$L_i = \{ x \in \mathbb{R}^d : H_{X,y}(x) \geq M_{H_{X,y}} \cdot 2^{-i} \} \,.$$

For any $x \in \mathbb{R}^d$ recall that

$$G_{X,y}(x) = M_{H_{X,y}} 2^{-\lfloor \log_2(M_{H_{X,y}}/H_{X,y}(x)) \rfloor} \,.$$

For any $A \subseteq \mathbb{R}^d$, let $\chi_A : \mathbb{R}^d \to \{0, 1\}$ be the indicator function for $A$. It is immediate that for all $x \in \mathbb{R}^d$,

$$G_{X,y}(x) = M_{H_{X,y}} \sum_{i=1}^{\infty} 2^{-i} \chi_{L_i}(x)$$

$$= M_{H_{X,y}} \sum_{i=1}^{m} 2^{-i} \chi_{L_i}(x) + 2^{-m} \chi_{L_i}(m) \qquad (\text{since } H_{X,y}(x) = 0 \text{ for all } x \notin L_m)$$

Let

$$c = \sum_{i=1}^{m} 2^{-i} \mathrm{vol}(L_i) + 2^{-m} \mathrm{vol}(L_m).$$

We have

$$\int_{\mathbb{R}^d} G_{X,y}(x) \mathrm{d}x = M_{H_{X,y}} \left( \sum_{i=1}^{m} 2^{-i} \mathrm{vol}(L_i) + 2^{-m} \mathrm{vol}(L_m) \right) = M_{H_{X,y}} c. \tag{B.2}$$

Let

$$\widehat{G}_{X,y}(x) = G_{X,y}(x)/(M_{H_{X,y}}c).$$

It follows by (B.2) that $\widehat{G}_{X,y}$ is a probability density function.

Let $D$ be the probability distribution on $\{1, \ldots, m\}$, where

$$\Pr_{I \sim D}[I = i] = \begin{cases} \operatorname{vol}(L_i) \cdot 2^{-i}/c & \text{if } i \in \{1, \ldots, m-1\} \\ 2 \cdot \operatorname{vol}(L_m) \cdot 2^{-m}/c & \text{if } i = m \end{cases}$$

For any $i \in [m]$, let $u_i$ be the uniform probability density function on $L_i$. To sample from $\widehat{G}_{X,y}$, we can first sample $I \sim D$, and then sample $Z \sim u_I$.

Recall that $\widehat{p}_{X,y} : \mathbb{R}^d \to \mathbb{R}_{\geq 0}$ is the probability density function obtained by normalizing $H_{X,y}$; that is, for all $x \in \mathbb{R}^d$ we have

$$p_{X,y}(x) = H_{X,y}(x)/c',$$

where

$$c' = \int_{\mathbb{R}^d} H_{X,y}(x)\mathrm{d}x.$$

Consider the following random experiment: first sample $Z \sim \widehat{G}_y$, and then accept with probability $H_{X,y}(Z)/G_{X,y}(Z)$; conditioning on accepting, the resulting random variable $Z \in \mathbb{R}^d$ is distributed according to $\widehat{H}_{X,y}$. Note that since for all $x \in \mathbb{R}^d$, $G_{X,y}(x)/2 \leq H_{X,y}(x) \leq G_{X,y}(x)$, it follows that we always accept with probability at least $1/2$. Let $\alpha$ be the probability of accepting. Then

$$\alpha = \int_{\mathbb{R}^d} \widehat{G}_{X,y}(x)(H_{X,y}(x)/G_{X,y}(x))\mathrm{d}x,$$

and thus

$$\begin{aligned} \int_{\mathbb{R}^d} H_{X,y}(x)\mathrm{d}x &= \int_{\mathbb{R}^d} G_{X,y}(x)(H_{X,y}(x)/G_{X,y}(x))\mathrm{d}x \\ &= M_{H_{X,y}}c \int_{\mathbb{R}^d} \widehat{G}_{X,y}(x)(H_{X,y}(x)/G_{X,y}(x))\mathrm{d}x \\ &= M_{H_{X,y}}c\alpha \,. \end{aligned} \tag{B.3}$$

By Theorem 2, for each $i \in [m]$, we compute an estimate, $\widetilde{\operatorname{vol}}(L_i)$, to $\operatorname{vol}(L_i)$, to within relative error $\delta$, using $\operatorname{poly}(d, 1/\delta, \log(1/\tau'))$ oracle calls, with probability at least $\tau'$, where $\tau' = \tau/n^b$, for some constant $b > 0$ to be determined; moreover, by Theorem 3, we can efficiently sample, using $\operatorname{poly}(d, 1/\delta)$ oracle calls, from a probability distribution $\widetilde{u}_i$ with $\|u_i - \widetilde{u}_i\| \leq \delta$. Each of these oracle calls is a membership query in some superlevel set of $H_{X,y}$. This membership query can clearly be implemented if we can compute that value $H_y$ at the desired query point $x$, which can be done in time $\operatorname{poly}(n, d)$ using LP (3.4). Thus, each oracle call takes time $\operatorname{poly}(n, d)$. Let

$$\widetilde{c} = \sum_{i=1}^{m} 2^{-i}\widetilde{\operatorname{vol}}(L_i) + 2^{-m}\widetilde{\operatorname{vol}}(L_m). \tag{B.4}$$

Since for all $i \in [m]$, $\operatorname{vol}(L_i)/(1+\delta) \leq \widetilde{\operatorname{vol}}(L_i) \leq \operatorname{vol}(L_i)(1+\delta)$, it is immediate that

$$c/(1+\delta) \leq \widetilde{c} \leq c(1+\delta) \,.$$

Recall that Algorithm 2 uses the probability distribution $\widetilde{D}$ on $[m]$, where

$$\Pr_{I \sim \widetilde{D}}[I = i] = \begin{cases} \widetilde{\operatorname{vol}}(L_i) \cdot 2^{-i}/\widetilde{c} & \text{if } i \in \{1, \ldots, m-1\} \\ 2 \cdot \widetilde{\operatorname{vol}}(L_m) \cdot 2^{-m}/\widetilde{c} & \text{if } i = m \end{cases}$$

Consider the following random experiment, which corresponds to Steps 5–6 of Algorithm 2: We first sample $I \sim \widetilde{D}$, and then we sample $Z \sim \widetilde{u}_I$. The resulting random vector $Z \in \mathbb{R}^d$ is distributed according to

$$\widetilde{G}_{X,y}(x) = \frac{1}{\widetilde{c}} \left( \sum_{i=1}^{m} 2^{-i}\widetilde{\operatorname{vol}}(L_i)\widetilde{u}_i(x) + 2^{-m}\widetilde{\operatorname{vol}}(L_m)\widetilde{u}_m(x) \right).$$

Next, consider the following random experiment, which captures Steps 5–8 of Algorithm 2: We sample $Z \sim \widetilde{G}_{X,y}$, and we accept with probability $H_{X,y}(Z)/G_{X,y}(Z)$. Let $\widetilde{H}_{X,y}$ be the resulting probability density function supported on $\mathbb{R}^d$ obtained by conditioning the above random experiment on accepting. Let $\widetilde{\alpha}$ be the acceptance probability. We have

$$\widetilde{\alpha} = \int_{\mathbb{R}^d} (H_{X,y}(x)/G_{X,y}(x))\widetilde{G}(x)\mathrm{d}x.$$

We have

$$\|D_i - \widetilde{D}_i\|_1 = \sum_{i=1}^{m-1} 2^{-i} \cdot \left| \frac{\mathrm{vol}(L_i)}{c} - \frac{\widetilde{\mathrm{vol}(L_i)}}{\widetilde{c}} \right| + 2 \cdot 2^{-m} \cdot \left| \frac{\mathrm{vol}(L_m)}{c} - \frac{\widetilde{\mathrm{vol}(L_m)}}{\widetilde{c}} \right|$$

$$= \sum_{i=1}^{m-1} 2^{-i} \cdot \left| \frac{\mathrm{vol}(L_i)}{c} - \frac{\mathrm{vol}(L_i)(1+\delta)}{c/(1+\delta)} \right| + 2 \cdot 2^{-m} \cdot \left| \frac{\mathrm{vol}(L_m)}{c} - \frac{\mathrm{vol}(L_m)(1+\delta)}{c/(1+\delta)} \right|$$

$$\leq \sum_{i=1}^{m-1} 2^{-i} \frac{\mathrm{vol}(L_i)}{c} 3\delta + 2 \cdot 2^m \frac{\mathrm{vol}(L_m)}{c} 3\delta$$

$$= 3\delta.$$

It follows that

$$\|\widehat{G}_{X,y} - \widetilde{G}_{X,y}\|_1 \leq \|D_i - \widetilde{D}_i\| + \max_i \|u_i - \widetilde{u}_i\|_1 \leq 3\delta + \delta \leq 4\delta,$$

and so

$$|\alpha - \widetilde{\alpha}| \leq \int_{\mathbb{R}^d} \frac{H_{X,y}(x)}{G_{X,y}(x)} \left| \widehat{G}_{X,y}(x) - \widetilde{G}_{X,y}(x) \right| \mathrm{d}x \leq \int_{\mathbb{R}^d} \left| \widehat{G}_{X,y}(x) - \widetilde{G}_{X,y}(x) \right| \mathrm{d}x \leq \|\widehat{G}_{X,y} - \widetilde{G}_{X,y}\|_1 \leq 4\delta.$$

Note that $p_{X,y}(x)/\alpha = \widehat{G}_{X,y}(x)\frac{H_{X,y}(x)}{G_{X,y}(x)}$ and $\widetilde{H}_{X,y}(x)/\widetilde{\alpha} = \widetilde{G}_{X,y}(x)\frac{H_{X,y}(x)}{G_{X,y}(x)}$ and so

$$\|\widetilde{H}_{X,y} - p_{X,y}\|_1 \leq \alpha \left( \|\widetilde{H}_{X,y}/\alpha - p_{X,y}/\alpha\|_1 + \|p_{X,y}/\widetilde{\alpha} - p_{X,y}/\alpha\|_1 \right) \qquad \text{(by the triangle inequality)}$$

$$= \alpha \left( \|\widetilde{H}_{X,y}/\alpha - p_{X,y}/\alpha\|_1 + |1/\widetilde{\alpha} - 1/\alpha| \right)$$

$$= \alpha \int_{\mathbb{R}^d} (H_{X,y}(x)/G_{X,y}(x))|\widetilde{G}_{X,y}(x) - p_{X,y}(x)| + |\alpha - \widetilde{\alpha}|/\widetilde{\alpha}$$

$$\leq \|p_{X,y} - \widetilde{G}_{X,y}\|_1 + 2|\alpha - \widetilde{\alpha}|$$

$$\leq 12\delta,$$

which establishes that the random vector $Z$ that Algorithm 2 outputs is distributed according to a probability distribution $\widetilde{\phi}$ such that $\|\widetilde{\phi} - p_{X,y}\|_1 \leq 10\delta$, as required.

In order to bound the running time, we observe that all the steps of the algorithm can be implemented in time $\mathrm{poly}(n, d, \|y\|_\infty, 1/\delta, \log(1/\tau))$. The most expensive operation is approximating the volume of a superlevel set $L_i$ and sampling for $L_i$, using Theorems 2 and 3. By the above discussion, using LP (3.4) and Lemma 4 each of these operations can be implemented in time $\mathrm{poly}(n, d, 1/\delta, \log(1/\tau))$. The algorithm succeeds if all the invocations of the algorithm of Theorem 2 are successful; by the union bound, this happens with probability at least $1 - \tau'\mathrm{poly}(n) = 1 - \tau'n^b\mathrm{poly}(n) \geq 1 - \tau$, where the inequality follows by choosing some sufficiently large constant $b > 0$. This establishes part (1) of the Lemma.

It remains to prove part (2). By (B.3) we have that $\gamma = M_{H_{X,y}}c\alpha$. Algorithm $\mathcal{A}_2$ proceeds as follows. First, we compute $M_{H_{X,y}}$. By the convexity of $h_{X,y}$, it follows that the maximum value of $M_{H_{X,y}}$ is attained on some sample point $x_i$; that is, $M_{H_{X,y}} = \max_{i \in [n]} H_{X,y}(x_i)$. Since we can evaluate $H_y$ in polynomial time using LP (3.4), it follows that we can also compute $M_{H_{X,y}}$ in polynomial time. Next, we compute $\widetilde{c}$ using formula B.4. Arguing as in part (1), this can be done in time $\mathrm{poly}(n, 1/\delta, \log(1/\tau))$, and with probability at least $1 - \tau/2$. Finally, we estimate $\widetilde{\alpha}$. The value of $\widetilde{\alpha}$ is precisely the acceptance probability of the random experiment described in Steps 5–8 of Algorithm 2. Since $\alpha \geq 1/2$, and $|\alpha - \widetilde{\alpha}| \leq 4\delta$, it follows that for $\delta < 1/16$, we can compute an estimate $\bar{\alpha}$ of the value of $\widetilde{\alpha}$, to within error $1 + O(\delta)$, with probability at least $1 - \tau/2$, after $O(\log(1/\tau))$ repetitions of the random experiment. The output of algorithm $\mathcal{A}_2$ is $\gamma' = M_{H_{X,y}}\widetilde{c}\bar{\alpha}$.

We obtain that, with probability at least $1 - \tau$, we have
$$\gamma' = M_{H_{X,y}} \widetilde{c}\bar{\alpha} \leq M_{H_{X,y}} c(1+\delta)\alpha(1+O(\delta)) = \gamma(1+O(\delta)) \, ,$$
and
$$\gamma' = M_{H_{X,y}} \widetilde{c}\bar{\alpha} \geq M_{H_{X,y}} (c/(1+\delta))(\alpha/(1+O(\delta))) = \gamma/(1+O(\delta)) \, ,$$
which concludes the proof. □

## C  Introduction To Exponential Families

In this section, we give a brief overview of exponential families that covers just the material necessary to appreciate the connection between exponential families and the log-concave maximum likelihood problem. We refer to [61] for a more complete treatment of exponential families.

An *exponential family* parameterized by $\theta \in \mathbb{R}^n$ with *sufficient statistic* $T(x)$, with carrier density $h$ measurable and non-negative is a family of probability distributions of the form
$$p_\theta(x) = \exp(\langle T(x), \theta \rangle - A(\theta))h(x).$$
The *log-partition* function $A(\theta)$ is defined to normalize the integral of the density
$$A(\theta) = \log \int \exp(\langle T(x), \theta \rangle)h(x)dx.$$

It makes sense to restrict our attention to values of $\theta$ that give a valid probability density. The set of *Canonical Parameters* $\Theta$ is defined such that $\Theta = \{\theta \mid A(\theta) < \infty\}$.

We say that an exponential family is *minimal* if $\theta_1 \neq \theta_2$ implies $p_{\theta_1} \neq p_{\theta_2}$. This is necessary and sufficient for statistical identifiability.

One reason exponential families are well studied is that we have an algorithm that computes the maximum likelihood estimate via a convex program.

The maximum likelihood parameters $\theta^\star$ for a set of iid samples $X_1, \ldots, X_n$ are:
$$\theta^\star = \arg\max_\theta \prod_i p_\theta(X_i) = \arg\max_\theta \log \prod_i p_\theta(X_i)$$
$$= \arg\max_\theta \sum_i \langle T(X_i), \theta \rangle - nA(\theta) - \sum_i \log h(x_i) = \arg\max_\theta \left\langle \frac{1}{n} \sum_i T(X_i), \theta \right\rangle - A(\theta)$$
(C.1)

We refer to the optimization in Equation (C.1) as the *exponential maximum likelihood optimization*. The last equation helps highlight why $T(x)$ is referred to as the sufficient statistic. No other information is needed about the data points to compute both the likelihood and the maximum likelihood estimator.

One reason why exponential families are important is that the geometry of the optimization in Equation (C.1) has several nice properties.

**Fact 3.** *$A(\theta)$ of exponential families satisfies the following properties: (a) $A(\theta) \in C^\infty$ on $\Theta$. (b) $A(\theta)$ is convex. (c) $\Delta A(\theta) = \mathbb{E}_{x \sim p(\theta)}[T(x)]$. (d) If the exponential family is minimal, $A(\theta)$ is strictly convex.*

Note that properties $(b), (c)$ are very similar to the definition of locally exponential families. The fact that tent distributions maintain some of these properties is exactly what enables the efficient algorithm in this paper.

### C.1  Analogy Between Log-Concave MLE and Exponential Family MLE

In the case of exponential families, at each time step, the algorithm maintains a distribution (from the hypothesis class) and generates a single sample from this distribution. The sufficient statistic of the exponential family can then be used to compute a subgradient. The computational efficiency follows from the convexity of the log-likelihood function, and existence of efficient samplers and procedures for computing the sufficient statistic. We portray this stochastic gradient method for

exponential families, together with the analogous form of our algorithm for log-concave distributions.

| **Exponential Family MLE** | **Log-Concave MLE** |
|---|---|
| Optimization Formulation: | Optimization Formulation: |

$$\max_y \langle \mu, y \rangle - \log \int \exp\left(\langle T(x), y \rangle\right) dx \qquad \max_y \langle \mathbb{1}, y \rangle - \log \int \exp\left(\langle T_{X,y}(x), y \rangle\right) dx$$

---

**Algorithm 3** Stochastic First Order Algorithm

   **function** COMPUTEEXPFAMMLE($X_1, \dots X_n$)
      $y \leftarrow y_{init}$
      **for** $i \leftarrow 1, m$ **do**
         $s \sim p(y)$                ▷ sample
         $y \leftarrow y + \eta_i \left(\mu - T(s)\right)$   ▷ subgradient
      **return** $y$

**Algorithm 4** Stochastic First Order Algorithm

   **function** COMPUTELOGCONMLE($X_1, \dots X_n$)
      $y \leftarrow 0$
      **for** $i \leftarrow 1, m$ **do**
         $s \sim p(X, y)$
         $y \leftarrow y + \eta_i \left(\frac{1}{n}\mathbb{1}_n - T_{X,y}(s)\right)$
      **return** $y$

---

# D  Learning Multivariate Log-Concave Densities

In this section, we combine our Theorem 1 with known sample complexity bounds to give the first computationally efficient and sample near-optimal proper learner for multivariate log-concave densities.

Recall that the squared Hellinger loss between two distributions with densities $f, g : \mathbb{R}^d \to \mathbb{R}_+$ is $h^2(f, g) = (1/2) \cdot \int_{\mathbb{R}^d} (\sqrt{f(x)} - \sqrt{g(x)})^2 dx$. Combined with the known rate of convergence of the log-concave MLE with respect to the squared Hellinger loss [16, 23], Theorem 1 implies the following:

**Theorem 4.** *Fix* $d \in \mathbb{Z}_+$ *and* $0 < \epsilon, \tau < 1$. *Let* $n = \tilde{\Omega}\left((d^2/\epsilon)\ln(1/\tau)\right)^{(d+1)/2}$. *There is an algorithm that, given $n$ iid samples from an unknown log-concave density $f_0 \in \mathcal{F}_d$, runs in $\mathrm{poly}(n)$ time and outputs a log-concave density $h^* \in \mathcal{F}_d$ such that with probability at least $1 - \tau$, we have that $h^2(h^*, f_0) \le \epsilon$.*

We note that Theorem 4 yields the first efficient proper learning algorithm for multivariate log-concave densities under a global loss function. The proof follows by combining Theorem 1 with the following lemma:

**Lemma 5.** *Let* $n = \Omega_d \left((1/\epsilon)\ln(1/(\epsilon\tau))\right)^{(d+1)/2}$. *Let* $\widehat{f_n}$ *be the MLE of $n$ samples drawn from $f_0 \in \mathcal{F}_d$. Let $h^*$ be a log-concave density that is supported on the convex hull of the samples with $\ell(h^*) \ge \ell(\widehat{f_n}) - \epsilon/16$. Then with probability at least $1 - \tau$ over the samples, $h^2(h^*, f_0) \le \epsilon$.*

We write $f_n$ for the empirical density over the samples $X_1, \dots, X_n$. The proof is a minor modification of the arguments in Section 3 of [16], using the following lemma [23]:

**Lemma 6** (Theorem 4 from [23]). *For any $t > 0$, we have except with probability $2\exp(-2t^2)$ that for any convex set $C$,*

$$|f_n(C) - f_0(C)| \le O_d(n^{-2/(d+1)}) + t/\sqrt{n}.$$

*Proof.* The proof follows Section 3 of [16], except that we need to replace Lemma 10 of that paper with Lemma 6 and that we use $h^*$ in place of $\widehat{f_n}$. We will sketch the proof here and highlight the modified components of that proof.

Lemma 10 of [16] had that, except with probability $\tau/3$, for all convex sets $C$, $|f_n(C) - f_0(C)| \le \epsilon/32 \ln(100n^4/\tau^2)$. We take $n = \Omega_d \left((1/\epsilon)\ln(1/(\epsilon\tau))\right)^{(d+1)/2}$ and $t = \sqrt{\ln(6/\tau)/2}$ in Lemma 6 and so $n^{-2/(d+1)} = O_d(\epsilon/\ln(1/\epsilon\tau)) = O_d(\epsilon/\ln(n/\tau))$ and $t/\sqrt{n} \le \sqrt{\ln(\tau)}(\epsilon/(\ln(\epsilon\tau)))^{-(d+1)/2} \le O(\epsilon/\ln(n/\tau))$ for $d \ge 2$. With a sufficiently large constant in the $\Omega_d$, we obtain that $|f_n(C) - f_0(C)| \le \epsilon/K \ln(100n^4/\tau^2)$ except with probability $\tau/3$ where $K$ is a constant large enough to make the subsequent proof work.

This gives the improved sample complexity. We now need to argue that replacing $\widehat{f_n}$ with $h^*$ does not affect the proof.

Corollary 9 of [16] gave that except with probability $\tau/10$, all samples lie in a set $S$, which is the set where $f_0(x) \geq p_{\min}$ for $p_{\min} = M_{f_0}/(n^4 100/\tau^2)$, where we use the notation $M_f$ for the maximum value of a density $f$.. When this holds both $\widehat{f}_n$ and $h^*$ are supported on $S$. Examination of the proof of Lemma 18 from [16] shows that we can relax the inequality $\ell(f) \leq \ell(f_0)$ to $\ell(f) \leq \ell(f_0) - \epsilon/16$ for any $f$ with maximum value $M_f$ has $M_f = \Omega(\ln(100n^4/\tau^2))$. In partuclar, since $\ell(h^*) \geq \ell(\widehat{f}_n)) - \epsilon/16 \geq \ell(f_0) - \epsilon/16$, we have $M_{h^*} = O(\ln(100n^4/\tau^2))$.

Then we define $g_h(x)$ supported on $S$ as the normalisation of $\max\{p_{\min}, h^*(x)\}$ for $x \in S$. The proof of Lemma 17 in [16] required only that $\widehat{f}_n$ is supported on $S$ and so we can obtain the same result for $g_h$ and $h^*(x)\}$ i.e. that $g_h(x) = \alpha \max\{p_{\min}, h^*(x)\}$ for $1 - \epsilon/32 \leq \alpha \leq 1$ and that the total variation distance is small,

$$d_{\mathrm{TV}}(g_h, h^*) \leq 3\epsilon/64 . \tag{D.1}$$

Note that since the superlevel sets of $\ln \max\{p_{\min}, h^*(x)\}$ are convex, we can use our application of Lemma 6 to bound the error in it's expectation as

$$|\mathbb{E}_{X \sim f_0}[1_S \ln(\max\{h^*(X), p_{\min}\})] - \mathbb{E}_{X \sim f_n}[1_S \ln(\max\{h^*(X), p_{\min}\})]| \leq (M_{h^*} - p_{\min})\epsilon/K \ln(100n^4/\tau^2)$$
$$\leq \epsilon/4 \tag{D.2}$$

for large enough $K$.

We now follow the proof of Lemma 19 in [16]. We have that

$$\mathbb{E}_{X \sim f_0}[\ln g_h(X)] = \mathbb{E}_{X \sim f_0}[1_S(x) \ln(\alpha \max\{p_{\min}, h^*(x)\})]$$
$$\geq \mathbb{E}_{X \sim f_0}[1_S(x) \ln \max\{p_{\min}, h^*(x)\}] - \epsilon/16 \qquad \text{(since } a > 1 - \epsilon/32)$$
$$\geq \mathbb{E}_{X \sim f_0}[1_S \ln(\max\{h^*(X), p_{\min}\})] - \epsilon/16$$
$$\geq \mathbb{E}_{X \sim f_n}[1_S \ln(\max\{h^*(X), p_{\min}\})] - 3\epsilon/16 \qquad \text{by (D.2)}$$
$$\geq \frac{1}{n} \sum_i \ln h^*(X_i) - 3\epsilon/16$$
$$\geq \frac{1}{n} \sum_i \ln \widehat{f}_n(X_i) - \epsilon/4$$
$$\geq \frac{1}{n} \sum_i \ln f_0(X_i) - \epsilon/4$$
$$\geq \mathbb{E}_{X \sim f_0}[\ln f_0(X)] - 3\epsilon/8. \qquad \text{(using Lemma 14 of [16])} \tag{D.3}$$

Thus, we obtain that

$$\mathrm{KL}(f_0||g) = \mathbb{E}_{X \sim f_0}[\ln f_0(X)] - \mathbb{E}_{X \sim f_0}[\ln g_h(X)] \leq 3\epsilon/8. \tag{D.4}$$

For the next derivation, we use that the Hellinger distance is related to the total variation distance and the Kullback-Leibler divergence in the following way: For probability functions $k_1, k_2 : \mathbb{R}^d \to \mathbb{R}$, we have that $h^2(k_1, k_2) \leq d_{\mathrm{TV}}(k_1, k_2)$ and $h^2(k_1, k_2) \leq \mathrm{KL}(k_1||k_2)$. Therefore, we have that

$$h(f_0, h^*) \leq h(f_0, g_h) + h(g_h, h^*)$$
$$\leq \mathrm{KL}(f_0||g_h)^{1/2} + d_{\mathrm{TV}}(g_h, h^*)^{1/2}$$
$$= (3\epsilon/8)^{1/2} + (3\epsilon/64)^{1/2} \qquad \text{(by (D.4) and (D.1))}$$
$$\leq \epsilon^{1/2} ,$$

concluding the proof. $\qquad\qquad\square$