[Reviews · NeurIPS 2019]

Reviewer 1



In terms of related work: are there any results on approximating a function by a convex function -- that seems like a simpler question and looks somewhat relevant. It would help to give some one/two sentence intuition for the definition of the tent function. In the related work you mention a lower bound of (1/eps)^O(d) for *any* estimator -- I didn't quite understand the exact difference from your formulation that allows a poly time upper bound. I understand that there is much prior work on this problem and the result seems interesting and significant -- but it would be nice to point out any specific applications or directions that could potentially be enabled by this finding.

Reviewer 2



The paper proposes the first polynomial time algorithm for computing the MLE of log-concave distributions in high dimensions. The proposed approach leverages the fact that log-concave MLE is contained in “tent densities”. Based on this observation, they propose a stochastic convex optimization algorithm for computing MLE. The main subroutine of the algorithm is an algorithm to sample from tent densities. The paper is theoretically interesting and provides a polynomial time algorithm, however, the degree of the polynomial is quite high. For example, step 1 of the Algorithm 2 takes time O(d^5) for d dimensional data. Further, given the sample complexity of the algorithm itself is exponential in d, the advantage of the polynomial time algorithm is not clear. In lemma 1, tent poles are defined as points X_i, but in line 184 it is defined as pairs (X_i, y_i) Is the epsilon in the sample complexity discussion (lines 156 - 170) different than the epsilon in Definition 2? If so, please clarify. Update: Thanks for answering my questions.

Reviewer 3



Post-rebuttal: The authors have promised to incorporate an exposition of the sampler in the revised paper, I believe that will make the paper a more self-contained read. I maintain my rating of strong accept (8). --------------------------------- The family of log-concave distributions is a powerful and general family of distributions that are often used in machine learning (they include most of the common distribution families). I think this paper makes very nice contributions to the fundamental question of estimating the MLE distribution given a bunch of observations. I think the key contributions can be broken up into two key parts: - A bunch of simple but elegant structural results for the MLE distribution in terms of 'tent distributions' -- distributions such that its log-density is piecewise linear, and is supported over subdivisions of the convex hull of the datapoints. This allows them to write a convex program for optimizing over tent distributions. This program captures the MLE distribution. - In order to apply SGD (or other optimization methods) to optimize this program, we need a (sub)gradient oracle. Constructing such an oracle is the other key contribution of this paper. This part is technically involved and the authors build an algorithm to approximately sample from the current distribution. The authors give the first efficient algorithm for MLE estimation problem over a natural and important class of distributions. I think the paper merits acceptance on the strength of the results, and the techniques necessary.

[Author Response · NeurIPS 2019]

First we would like to thank all the reviewers for their feedback. We address the reviewers' questions below.

**Reviewer 1:** "In the related work you mention a lower bound of $(1/\epsilon)^{O(d)}$ for *any* estimator – I didn't quite understand the exact difference from your formulation that allows a poly time upper bound."

The $(1/\epsilon)^{O(d)}$ is a bound on the number of independent samples, $n$, required to learn a $d$-dimensional logconcave distribution within error $\epsilon$ in squared Hellinger distance, and this bound is tight in the worst-case. Our algorithm runs in time polynomial in $d$, $n$, and 1/error, and computes the logconcave MLE of *any* set of $n$ points in $\mathbf{R}^d$. The best previous algorithms had runtime polynomial in $n^{\Omega(d)}$.

"I understand that there is much prior work on this problem and the result seems interesting and significant – but it would be nice to point out any specific applications or directions that could potentially be enabled by this finding."

Any application where one might be tempted to model the distribution as a multivariate Gaussian, may also be suitable for the log-concave MLE. Provided sufficient data, the log-concave MLE would capture properties such as asymmetry and skewness in the distribution. For example, this has been used to better predict breast cancer malignancies [1].

**Reviewer 3:** "The paper is theoretically interesting and provides a polynomial time algorithm, however, the degree of the polynomial is quite high. For example, step 1 of the Algorithm 2 takes time $O(d^5)$ for d dimensional data. Further, given the sample complexity of the algorithm itself is exponential in d, the advantage of the polynomial time algorithm is not clear."

An interesting direction of future work is reducing the $d$-dependence. The main bottleneck of the current algorithm is the volume computation (step 1 of Algorithm 2). Recent developments in RHMC based methods for volume computation have resulted in much faster algorithms for computing volumes of polytopes [2]. However, the aforementioned algorithms require a different oracle model and do not apply directly in our setting. That said, we are optimistic that similar ideas might apply, and could plausibly lead to a much more efficient implementation of our algorithm.

"In lemma 1, tent poles are defined as points $X_i$, but in line 184 it is defined as pairs $(X_i, y_i)$"

Thank you for pointing this out. It will be fixed in the next revision.

"Is the epsilon in the sample complexity discussion (lines 156 - 170) different than the epsilon in Definition 2? If so, please clarify."

Appendix $E$ shows that a small value of one implies a small value of the other. Thus, if suffices to think of them as equivalent in the regime where the log-concave MLE has converged.

**Reviewer 5:**

"The authors could present some more exposition of the sampler."

We will add further exposition in the revised version of our paper.

# References

[1] Madeleine Cule, Richard Samworth, and Michael Stewart. Maximum likelihood estimation of a multi-dimensional log-concave density. *Journal of the Royal Statistical Society: Series B (Statistical Methodology)*, 72(5):545–607, 2010.

[2] Yin Tat Lee and Santosh S Vempala. Convergence rate of riemannian hamiltonian monte carlo and faster polytope volume computation. In *Proceedings of the 50th Annual ACM SIGACT Symposium on Theory of Computing*, pages 1115–1121. ACM, 2018.


[Meta-Review · NeurIPS 2019]

The submission provides a polynomial-time approximation algorithm for finding the maximum-likelihood log-concave density for a given set of data points in R^d, for arbitrary d. The work is theoretical in nature, with proofs and no experiments. The problem is very interesting, since log-concave distributions include may of the commonly used parametric families (such as Gaussian), and the log-concave MLE has also other interesting properties. Previously the sample-complexity of learning a log-concave distribution has been studied, but a polynomial-time algorithm has been lacking. The present work provides such an algorithm. The derivation of the algorithm requires solving several interesting and non-trivial technical sub-problems. The presentation is generally very good. On slightly negative side, the practical impact of the result in very high-dimensional settings might be limited because of the high degree of the polynomial dependence on d.